# Response of *Bromus valdivianus* (Pasture Brome) Growth and Physiology to Defoliation Frequency Based on Leaf Stage Development

Iván P. Ordóñez [1,2,3], Ignacio F. López [1,3,*], Peter D. Kemp [1,3], Daniel J. Donaghy [1], Yongmei Zhang [4] and Pauline Herrmann [5]

1 School of Agriculture and Environment, Massey University, Private Bag 11-222, Palmerston North 4440, New Zealand; ivan.ordonez@inia.cl (I.P.O.); P.Kemp@massey.ac.nz (P.D.K.); D.J.Donaghy@massey.ac.nz (D.J.D.)
2 Instituto de Investigaciones Agropecuarias, INIA, Kampenaike, Punta Arenas 6212707, Chile
3 Centro de Investigación en Suelos Volcánicos, Universidad Austral de Chile, Valdivia 5091000, Chile
4 Gansu Provincial Key Laboratory of Aridland Crop Science, Gansu Agricultural University, Lanzhou 730070, China; zhangyongm@gsau.edu.cn
5 Ecole Nationale Supérieure d'Agronomie et des Industries Alimentaires, ENSAIA, 54505 Nancy, France; pauline.herrmann56@gmail.com
* Correspondence: I.F.Lopez@massey.ac.nz

**Abstract:** The increase in drought events due to climate change have enhanced the relevance of species with greater tolerance or avoidance traits to water restriction periods, such as *Bromus valdivianus* Phil. (*B. valdivianus*). In southern Chile, *B. valdivianus* and *Lolium perenne* L. (*L. perenne*) coexist; however, the pasture defoliation criterion is based on the physiological growth and development of *L. perenne*. It is hypothesised that *B. valdivianus* needs a lower defoliation frequency than *L. perenne* to enhance its regrowth and energy reserves. Defoliation frequencies tested were based on *B. valdivianus* leaf stage 2 (LS-2), leaf stage 3 (LS-3), leaf stage 4 (LS-4) and leaf stage 5 (LS-5). The leaf stage development of *Lolium perenne* was monitored and contrasted with that of *B. valdivianus*. The study was conducted in a glasshouse and used a randomised complete block design. For *Bromus valdivianus*, the lamina length, photosynthetic rate, stomatal conductance, tiller number per plant, leaf area, leaf weights, root growth rate, water-soluble carbohydrates (WSCs) and starch were evaluated. *Bromus valdivianus* maintained six live leaves with three leaves growing simultaneously. When an individual tiller started developing its seventh leaf, senescence began for the second leaf (the first relevant leaf for photosynthesis). Plant herbage mass, the root growth rate and tiller growth were maximised at LS-4 onwards. The highest leaf elongation rate, evaluated through the slope of the lamina elongation curve of a fully expanded leaf, was verified at LS-4. The water-soluble carbohydrates (WSCs) increased at LS-5; however, no statistical differences were found in LS-4. The LS-3 and LS-2 treatments showed a detrimental effect on WSCs and regrowth. The leaf photosynthetic rate and stomatal conductance diminished while the leaf age increased. In conclusion, *B. valdivianus* is a 'six-leaf' species with leaf senescence beginning at LS-4.25. Defoliation at LS-4 and LS-5 was optimum for plant regrowth, maximising the aboveground plant parameters and total WSC accumulation. The LS-4 for *B. valdivianus* was equivalent to LS-3.5 for *L. perenne*. No differences related to tiller population in *B. valdivianus* were found in the different defoliation frequencies.

**Keywords:** defoliation criterion; energy reserves; growing degree days; photosynthetic rate; stomatal conductance

## 1. Introduction

In temperate climates, a decline in rainfall and a concomitant increase in temperature during summer cause an increase in the soil matric potential [1], which can be exacerbated

by climate change [2,3]. The consequences are the diminishment pasture growth rate [4], species survival and of pasture persistence [5,6].

During a summer drought period, the soil matric potential in the first 20 cm closest to the surface can reach values greater than 1.543 MPa, which corresponds to the permanent wilting point [7]. Under these conditions, the survival and growth of shallow-rooted grasses, for example, *L. perenne*, can be compromised [7,8]. Plants with longer roots can access water in deep soil layers, increasing their capacity to avoid drought [5] and their resilience after drought [9]; consequently, they have higher growth rates than shallow-rooted species [10].

Drought-avoidance species (e.g., deep rooting species such as *Bromus valdivianus* Phil.) [11] can be incorporated into pastures to mitigate the decline in pasture growth under dry soil conditions [12]. *Bromus valdivianus*, a perennial grass species, with similar annual accumulated herbage mass and nutritive value to *L. perenne*, dominates pastures in non-irrigated fertile Andisol soils of southern Chile, often coexisting with *L. perenne*, *Holcus lanatus* L. and *Poa pratensis* L. [13]. It has been reported that *B. valdivianus*, compared to *L. perenne*, has tillers that are twice as heavy, with almost four times greater leaf area per tiller, a longer total lamina length per tiller, but only half the number of tillers per plant [13,14]. The high percentage of both species within the same pasture, measured by botanical composition [1,15], suggests that there is functional compatibility between *L. perenne* and *B valdivianus*. The different growth strategies of the two species [13,14] probably favour the persistence of *B. valdivianus* during soil water restriction, as it can obtain water from deep soil layers to continue growing [1,7,11,13].

In order to have a defoliation regime that favours the fast-growing species within a pasture [15], it is relevant to determine defoliation targets based on physiological and morphological principles. Leaf stage development in grasses (sequential leaf appearances and growth) has been indicated as a precise indicator for optimal defoliation time [16–20], because it is based on leaf appearance, a physiological trait, that is mainly influenced by temperature [21,22]. A higher defoliation frequency leads to a depletion of regrowth, aboveground herbage mass [16,23], root growth [24,25] and persistence of the plants [16,25]. Both are a consequence of the steady diminishment of the energy reserves in the storage organs (i.e., water-soluble carbohydrates) [16,25,26]. For *B. valdivianus*, there are no studies related to its defoliation frequency based on leaf stage development, just a few recent approximations [20,24]. However, the determination of the exact defoliation moment when the regrowth and energy reserves accumulation are encouraged rather than depressed has not yet been studied. In addition to the regrowth stage that encourages the accumulation of the energy reserves [16], the quantity of water-soluble carbohydrates (WSCs) in the storage organs is another important variable that can affect plant survival during drought periods [27].

In a pasture with a greater contribution of *B. valdivianus*, either as a monoculture or coexisting along with other grasses such as *L. perenne* [13], the defoliation criterion has been based on *L. perenne* research, targeting defoliation between the two- and three-leaf *L. perenne* regrowth stage, with the two-leaf stage representing the point at which the plant energy reserves are replenished (down limit) and the three-leaf stage (upper limit) representing the point at which leaf senescence begins [16,25,26,28]. However, the optimal defoliation for *B. valdivianus* may differ from that of *L. perenne*, which may negatively impact *B. valdivianus* pasture production, persistence and survival.

The aim was to determine the effects that defoliation at different stages of vegetative development have on *B. valdivianus* leaf elongation, below- and aboveground herbage mass and WSC and starch accumulation in the stubble. To accomplish this objective, the concentration and content of WSCs and starch in the tiller base (bottom 5 cm), along with lamina length for each tiller leaf, leaf number, tiller population, herbage mass, root mass, plant leaf mass and tiller weight were determined under different stages of leaf regrowth (LS) development (expansion of sequential new leaves per tiller). Furthermore,

the photosynthetic rate ($A_n$) and stomatal conductance ($G_s$) of *B. valdivianus* were assessed on leaves of different ages.

## 2. Materials and Methods

### 2.1. Experimental Conditions and Treatments Description

The study was conducted in a glasshouse at Massey University's Plant Growth Unit, Palmerston North, New Zealand. It was set up at the end of June 2016 and the treatments were implemented between 28 August 2016 and 25 January 2017.

The temperature in the glasshouse was evaluated at a 1 h frequency at the same height level as the pots. Extreme temperatures were prevented by using the two temperature control systems of the glasshouse, whereby temperatures higher than 25 °C automatically activated the opening of the glasshouse windows and the functioning of the ventilation system.

A total of 68 pots with 10 L volume capacity were used. The pot substrate was a mix of 50% soil (Manawatu silt loam) and 50% sand. All the pots were automatically irrigated 3 times per day with a tube system, meeting the water requirements of all plants by keeping the pots at field capacity (25% of the volumetric soil water content).

The plants were fertilised to maintain an adequate nutrient status; a short-term fertiliser was added at a rate of 100 g per $pot^{-1}$ with a concentration of 15% nitrogen (N), 6% phosphorus (P), 11.6% potassium (K), 1.5% magnesium (Mg), 6% sulphur (S), 2% iron (Fe) and 0.5% manganese (Mn), along with a long-term fertiliser at a rate of 100 g per $pot^{-1}$, which comprised 15% N, 2.2% P, 8.3% K, 0.2% Mg, 1.5% S, 1.5% Fe, 0.3% Mn and 0.2% zinc (Zn).

The study had 4 treatments and 17 blocks; however, only 6 blocks were used for all the continuous variable measurements, i.e., lamina length, herbage mass, leaf area, tiller population, net photosynthesis, stomatal conductance and root mass. The other 11 blocks were managed and grown under the same conditions as those 6 blocks and used only as extra blocks at the end of the study when a large amount of herbage material was required i.e., WSC and starch content evaluations. The pots within each block had similar light and humidity conditions. In order to avoid incident light and temperature differences between treatments, within blocks, the pots were randomly moved every two weeks.

In each individual pot, two seeds of *B. valdivianus* were sown at eight equidistance positions, where six were close to the edge and two were in the centre of each pot. After germination, only one seedling at each position was kept. The evaluated variables at plant and tiller levels were performed on the two plants located in the centre of the pots: Lamina length for each tiller leaf, number of leaves, leaf area per tiller, leaf mass per tiller, tiller number per plant and herbage mass per plant.

To stimulate tillering, the plants were defoliated three times before the study period began [29]. Each defoliation was performed when the plants reached an average height of 15 cm from the ground level. After the third defoliation, the 28 August 2016, the following treatments were applied: (1) Defoliation at leaf stage 2 (LS-2), (2) defoliation at leaf stage 3 (LS-3), (3) defoliation at leaf stage 4 (LS-4) and (4) defoliation at leaf stage 5 (LS-5).

For *L. perenne*, the "one-leaf stage" (1 LS) was classified as one fully expanded leaf (100%), the "one and a half leaf stage" (1.5 LS) was a fully expanded leaf (100%) plus a half extension of a second leaf (50%), the "two-leaf stage" (2 LS) was a total of two fully expanded leaves (100%), and so on [16,19]. However, unlike *L. perenne*, *B. valdivianus* expands three leaves simultaneously [14]. The leaves in expansion are a fraction of the fully expanded leaf size (100% of expansion or 1 LS). The scores used for those leaves in expansion were 0.25 LS (25% of expansion), 0.5 LS (50% of expansion) and 0.75 LS (75% of expansion). Two leaves warranted particular consideration: The older leaves (Leaf 1 and Leaf 2) showed limited growth because they were a remnant of actively growing leaves during the prior defoliation; Leaf 1 was not considered in determining leaf stage due to its limited growth after the prior defoliation, and Leaf 2 (leaf 2) grew until 75% or 0.75 LS.

As the present research was centred on *B. valdivianus* growth and development, the effect of defoliation frequency on *L. perenne* growth was not considered due to the existence of previous publications that had clarified the effect of the defoliation frequency on its growth and water-soluble carbohydrates content [16,19,25,26,28,30]. However, extra pots of *L. perenne* were grown under the same conditions as those of *B. valdivianus* and used as a known parameter for LS development. The defoliation of *L. perenne* was performed at the same time as the *B. valdivianus* LS-5 treatment.

When the LS-5 treatment occurred for the third time, the other treatments were sequentially defoliated according to their corresponding leaf stage (LS), and the study period ended. For this reason, every treatment had different total growing days. The final harvest began with LS-5 on 30 December 2016 (124 days of growth from the start of the study), followed by LS-2 on 5 January 2017 (130 days of growth), LS-3 on 7 January 2017 (132 days of growth) and LS-4 on 26 January 2017 (151 days of growth). The leaf expansion and all the frequency defoliation cycles were calculated based on the growing degree days (°C day; GDD) with a base temperature of 5 °C [21,22] based on previous studies performed with temperate climate grass species [31,32].

*2.2. Evaluated Variables*

At the beginning of the study period, three tillers in the centre of each plant (two plants giving a total of six marked tillers per pot) were marked by placing a thin, coloured wire at their base [31], and every two days, the leaf appearance, leaf number, lamina length and senescence were recorded for each tiller leaf. A new leaf was recorded as appearing when the tip of the new lamina was visible within the previous leaf sheath [21], and a leaf was considered fully expanded when its growth stopped and it reached its maximum length. The lamina length was measured as the distance between the lamina tip and collar section. For lamina length, only the green part of a lamina was considered, thus when a lamina began senescence, by decolouring from the tip to the ligule, the decoloured part of the lamina was not considered for lamina length measurements. In this way, it was possible to determine lamina senescence rate [33].

When a treatment reached the desired LS, the marked tillers were evaluated as it follows: First, they were cut to a height of 5 cm above ground level and individually evaluated for leaf area using an LI-3100C area meter (LI-COR, Nebraska, USA; accuracy of 1–5% using a resolution of 0.1 mm$^2$) and dry leaf mass, after drying in an oven at 70 °C for 48 h. Following this, the remainder of the plant containing the marked tillers (the two plants in the centre) was cut to a 5 cm height, herbage was collected, and dried in an oven at 70 °C for 48 h to obtain mass on a dry matter (DM) basis. The plants located close to the edge of the pots were also cut to a 5 cm height, and the herbage was discarded.

At the final harvest, the number of tillers/plants was recorded, and then the marked tillers were cut first to a 5 cm height as previously described, then to ground level (tiller base), and dried at 70 °C for 48 h. The 11 extra blocks ensured that the minimum amount of dry-matter herbage from tiller bases required was reached to measure WSCs and starch. Therefore, at the final harvest of the study, the tiller base of all non-marked tillers along with the tiller bases from the additional plants maintained in the 11 blocks were harvested to ground level. WSCs and starch were analysed using Gas–liquid chromatography [34]. The soil was washed from the roots to assess the root mass. Afterwards, the roots of the two centre plants were dried at 70 °C for 48 h and weighed.

The photosynthetic rate ($A_n$) and stomatal conductance ($G_s$) were evaluated in the LS-5 treatment using four fully expanded leaves and one growing leaf prior to the defoliation of the LS-5 treatment in order to assess the net and total photosynthesis of the species plus the effect of leaf age. For this evaluation, leaf 4 was the oldest leaf and leaf 8 was the youngest leaf. The evaluation used a LICOR-6400XT (LI-COR, Lincoln, NE, USA), with a 6400-02B LED light source chamber (LI-COR, Nebraska, USA). The photosynthetic photon flux density at the leaf surface was set at 1000 μmol photons, the leaf temperature was stabilised at 25 °C, a flow rate of 500 μmol s$^{-1}$ was set and the ambient $CO_2$ concentration

of the incoming gas stream was set at 400 μmol s$^{-1}$. The $A_n$ and $G_s$ were determined on two dates, 10 November 2016 and 22 December 2016, just before the harvests. For each of these variables, two leaves of the same age were set inside the chamber and leaf area was determined for each one using a digital calliper.

The effect of defoliation frequency on plant regrowth rates was assessed for the following parameters: Tiller population, lamina length, leaf area growth, shoot growth and root growth. For the last three parameters listed, accumulated growth was divided by the total growing days for each treatment as follow: LS-2 (130 days of growth), LS-3 (132 days of growth), LS-4 (151 days of growth) and LS-5 (124 days of growth). Leaf, shoot and root growth rates were calculated as follows: (1) Leaf elongation: The elongation rate was determined with the first fully expanded leaf of each treatment (leaf 3). For leaf elongation, quadratic equations were fitted. Moreover, calculation of the slope for all the equations was done considering 50 GDD, as this is the moment when WSCs have a greater influence on leaf regrowth [16,26]. (2) Considering the leaf area growth rate, the accumulated leaf area growth of all the defoliations was divided by the total of growing days for each treatment, determining the growing rate of the tiller leaf area (cm$^2$ day$^{-1}$). (3) For the shoot growth rate, the accumulated plant herbage mass harvested at each defoliation event (above 5 cm) was divided by the total growing days of each treatment, determining the shoot growth rate (g day$^{-1}$). (4) Concerning the root growth rate, the total root mass was divided by the total number of growing days of each treatment, estimating the root growth rate (g day$^{-1}$).

The phyllochron and the GDD required for a fully expanded leaf were estimated from the lamina length data collected every two days. An average between each leaf appearance was determined and the phyllochron was calculated. Leaf 3 was used to calculate the GDD necessary for a leaf to reach full expansion, as it was the first fully expanded leaf.

### 2.3. Experimental Design and Statistical Analysis

The study was set up as a randomised complete block design (4 treatments × 6 blocks). The statistical analyses were performed using SAS (version 9.4). The normality of the data was evaluated using the Shapiro–Wilk test ($p \leq 0.05$). Tiller weight and tiller leaf area were normalised with the natural logarithm function, and together with normal distributed data, such as phyllochron, leaf area growth rate, tiller leaf weight, $A_n$ and $G_s$, were analysed using an analysis of variance (ANOVA). Treatments effects were separated by applying Fisher's least significance difference test (LSD) ($p \leq 0.05$). It was not possible to normalise the tiller number, plant shoot growth rate, root growth rate and plant and tiller fructose, glucose and starch using function transformations, therefore non-parametric statistics were applied (box diagrams, mean, median, percentiles and standard deviation), and the Kruskal–Wallis test ($p \leq 0.05$) was used to compare the treatment medians. Canonical variate analysis (CVA) was performed to explain the differences between the treatments and their variation, based on the joint analysis of all the variables and data. CVA was undertaken on the standardised data, thus each variable had an average of zero and a standard deviation equal to one [35].

### 3. Results

#### 3.1. Bromus valdivianus Growth Dynamics

When the treatment on LS-5 reached the third defoliation, all the following treatments were consecutively harvested as they had achieved their corresponding LS development. LS-2 completed eight cycles, LS-3 finished six cycles, LS-4 accomplished five cycles and LS-5 completed three cycles (Figure 1).

*Bromus valdivianus* maintained six live leaves simultaneously: Three fully grown leaves (Figure 1, leaf 2, 3 and 4) and three actively growing leaves (Figures 1 and 2, leaf 5, 6 and 7). The first leaf (Leaf 1), indicated in Figure 1, is considered a 'residual leaf', being the last expanding leaf at the time of the prior defoliation event, and was not included in this analysis. Leaf 2 was a younger 'residual leaf' than Leaf 1 from the prior defoliation event; it

grew to 75% of the total length of a complete normal lamina, being given a score of 0.75 LS. The other two leaves that reached full expansion were leaf 3 and leaf 4, scored as 1 LS each.

When leaf 8 appeared, the oldest leaf (Leaf 2) started to senesce. As senescence began, the tillers had three completely expanded leaves, the second, third and fourth leaves (0.75 LS + 1LS + 1 LS), and three expanding leaves, the fifth, sixth and seventh leaves (0.75 LS, 0.5 LS and 0.25 LS). Thus, *B. valdivianus* leaf senescence started at 4.25 LS (Figure 1). The phyllochron was not affected by the defoliation frequency ($p > 0.05$), and on average, it took 96 °C d for one leaf to appear (Table 1). The leaves reached full expansion at 431 °C d GDD after leaf appearance, but with three leaves growing at the same time (Figure 1). Furthermore, for defoliation at the LS-5 treatment, the development of spikes and seeds was observed.

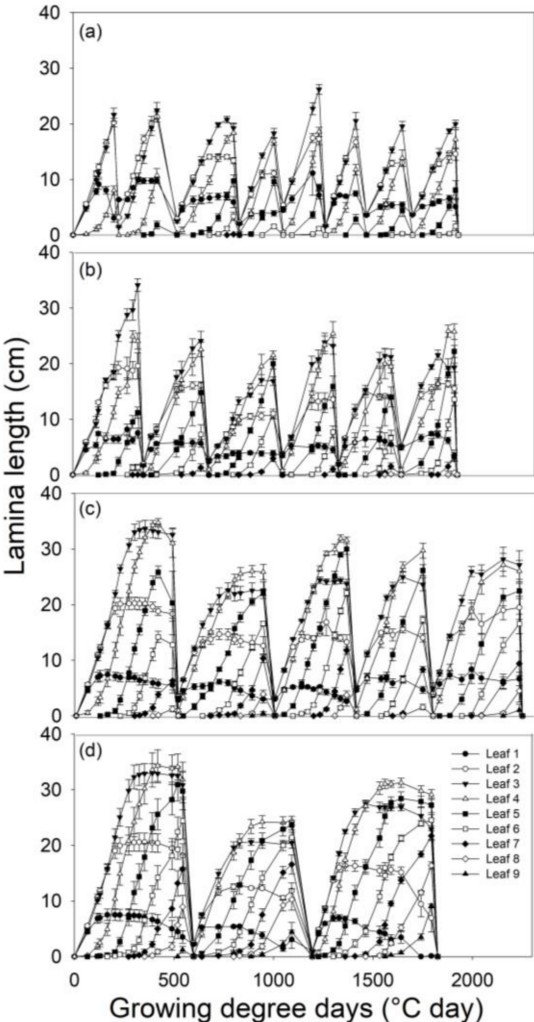

**Figure 1.** Tiller leaf growth dynamic and regrowth periods for each defoliation treatment for *B. valdivianus*. The defoliation frequencies were based on defoliation at (**a**) leaf stage 2 (LS-2); (**b**) leaf stage 3 (LS-3); (**c**) leaf stage 4 (LS-4); (**d**) leaf stage 5 (LS-5). Error bars: ±SEM (*n* = 6).

Figure 2 shows the differences between *B. valdivianus* and *L. perenne* in leaf stage development and leaf appearance over time, expressed as GDD. Furthermore, the GDD needed for a full expanded leaf can be extracted from Figure 2.

When defoliation was performed at LS-5, *B. valdivianus* presented nine leaves, and when its seventh leaf appeared, the second leaf started to senesce (Figure 1). *Lolium perenne*, under a frequency of defoliation that was dependent on *B. valdivianus* reaching LS-5, generated six leaves (Figure 2).

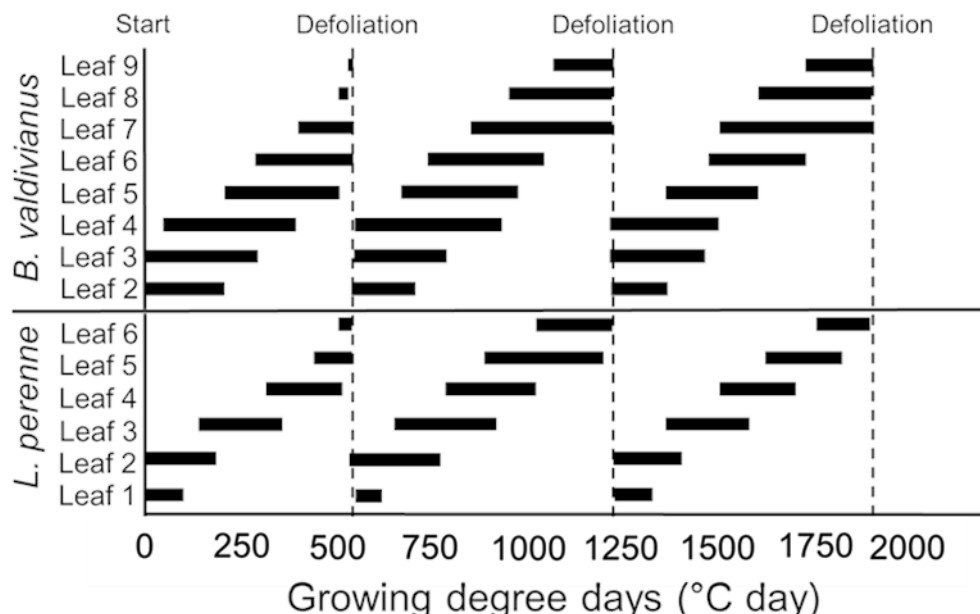

**Figure 2.** Scheme of the leaf expansion cycles based on GDD for the LS-5 treatment of *B. valdivianus* and the extra treatment of *L. perenne* based on the *B. valdivianus* (LS-5) defoliation frequency.

**Table 1.** Plant parameters at the final harvest and the accumulation over several defoliations intervals under different defoliation frequencies for *B. valdivianus* defoliated by leaf stage development (LS).

| Defoliation Treatments | Final Harvest Results [1] | | Sum of All Defoliations [2] | | |
| | Tiller Leaf Area (cm$^2$ tiller$^{-1}$) [3] | Tiller Weight (g tiller$^{-1}$) [4] | Tiller Leaf Weight (g tiller$^{-1}$) [5] | Tiller Leaf Area Growth Rate (cm$^2$ tiller$^{-1}$ day$^{-1}$) [6] | Phyllochron (°C Day) |
|---|---|---|---|---|---|
| LS-2 | 3.56 c (12.7 cm$^2$) | 0.24 d (0.134 g) | 0.11 d | 38.8 | 101.4 |
| LS-3 | 4.85 c (23.9 cm$^2$) | 0.33 c (0.25 g) | 0.22 c | 39.9 | 92.9 |
| LS-4 | 5.30 b (28.4 cm$^2$) | 0.49 b (0.48 g) | 0.30 b | 45.0 | 94.4 |
| LS-5 | 6.62 a (44.3 cm$^2$) | 0.57 a (0.60 g) | 0.35 a | 36.0 | 95.5 |
| *p*-value | *** | *** | *** | NS | NS |

The data in parenthesis are the values without normalisation. [1] Measurement was conducted during the final harvest, therefore is the accumulated effect over the parameters. [2] Sum of all defoliation is the accumulated outcome considering all defoliation of each treatment. [3] The tiller leaf area (sheath + lamina) considering only the final harvest of each treatment. [4] The whole tiller weight considering only the final harvest of each treatment. [5] Average tiller leaf weight (5 cm upwards) considering all defoliations of each treatment. [6] Tiller leaf area growth rate (5 cm upwards) considering the tiller leaf area during each defoliation for each treatment. *** means $p \leq 0.0001$; NS means no statistical differences.

The effects of defoliation frequency on plant and tiller growth were evaluated through the following variables:

(1)  Leaf elongation: Figure 3 shows the first fully expanded leaf (Leaf 3 in Figure 1). The growth rate of Leaf 3 was retarded at the beginning of its growth by the most frequent defoliation criterion (LS-2 and LS-3). The derivation of quadratic equations (Figure 3), replacing the X as GDD with 50 GDD (third day after a defoliation event), ordered the slopes as follow: LS-4 (0.1289) > LS-5 (0.1256) > LS-3 (0.115) > LS-2 (0.0945).

(2)  Leaf area growth rate: No statistical differences were found between the treatments ($p > 0.05$; Table 1).

(3)  Plant shoot growth rate: Statistical differences were determined ($p \leq 0.001$; Figure 4), where the results of shoot growth rate were as follows (mean $\pm$ sem): LS-5 0.23 g plant$^{-1}$ day$^{-1}$ ($\pm 0.017$ g) = LS-4 0.23 g plant$^{-1}$ day$^{-1}$ ($\pm 0.017$ g) > LS-3 0.18 g plant$^{-1}$ day$^{-1}$ ($\pm 0.013$ g) = LS-2 0.13 g plant$^{-1}$ day$^{-1}$ ($\pm 0.008$ g).

(4)  Root growth rate: Statistical differences were determined ($p \leq 0.001$; Figure 4), in which the root growth rate when the defoliation was at LS-5 was statistically higher than at LS-3 and at LS-2 but equal to LS-4. Root growth rates were as follows (mean ± sem): LS-5 0.049 g day$^{-1}$ plant$^{-1}$ (±0.007 g) = LS-4 0.018 g day$^{-1}$ plant$^{-1}$ (±0.003 g) and LS-4 = LS-3 0.011 g day$^{-1}$ plant$^{-1}$ (±0.002 g) = LS-2 0.008 g day$^{-1}$ plant$^{-1}$ (±0.001 g).

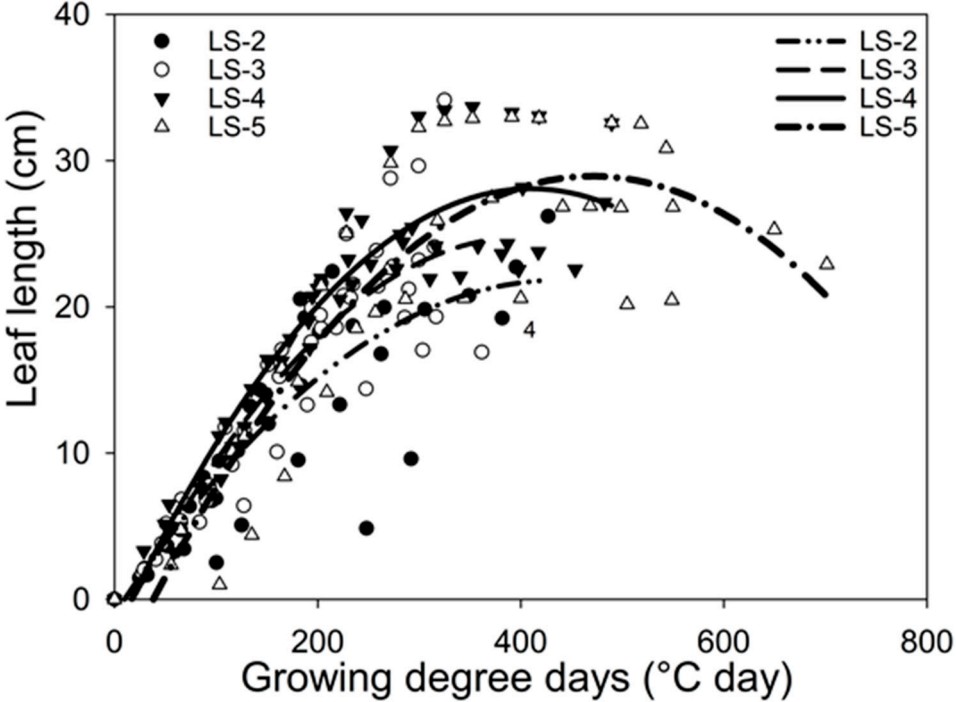

**Figure 3.** *Bromus valdivianus* growth rate of the first fully expanded lamina (Leaf 3) from each defoliation frequency, leaf stage 2 (filled circle; dotted with dashed line), leaf stage 3 (empty circle; dashed line), leaf stage 4 (filled triangle; continuous line) and leaf stage 5 (empty triangle; dotted line). The equations are for LS-2) y = −0.0001x$^2$ + 0.1045x − 1.0023 (R$^2$ = 0.71); LS-3) y = −0.0002x$^2$ + 0.135x − 2.3488 (R$^2$ = 0.83); LS-4) y = −0.0002x$^2$ + 0.1489x − 2.4398 (R$^2$ = 0.87); LS−5) y = −0.0002x$^2$ + 0.1456x − 5.3445 (R$^2$ = 0.80).

### 3.2. Non-Structural Carbohydrates Reserves

Defoliation frequency resulted in significant differences ($p \leq 0.001$) in the tiller and plant WSC content. Glucose and fructose decreased with more frequent defoliation (Figure 5). The median of the glucose content was as follows: LS-5 109.3 mg plant$^{-1}$ LS-4 56.3 mg plant$^{-1}$, LS-3 30.1 mg plant$^{-1}$ and LS-2 17.2 mg plant$^{-1}$. Moreover, the median of the fructose contents was as follows: LS-5 206.1 mg plant$^{-1}$, LS-4 113.2 mg plant$^{-1}$, LS-3 42.06 mg plant$^{-1}$ and LS-2 26.7 mg plant$^{-1}$. The median of the starch content followed a different pattern than glucose and fructose: LS-4 11.81 mg plant$^{-1}$, LS-5 8.95 mg plant$^{-1}$, LS-3 3.00 mg plant$^{-1}$ and LS-2 2.57 mg plant$^{-1}$. The median of the total WSC content followed a similar pattern to the glucose and fructose: LS-5 329 mg plant$^{-1}$, LS-4 209 mg plant$^{-1}$, LS-3 82 mg plant$^{-1}$ and LS-2 50 mg plant$^{-1}$.

As a general trend, the treatments LS-5 and LS-4 had the same content of WSCs and starch per tiller and per plant. The total WSCs showed that the LS-5 and LS-4 treatments were statistically similar. Such a result was also obtained between LS-4 and LS-3. However, LS-3 was lower than LS-5, and similar to LS-2.

### 3.3. Net Photosynthesis and Stomatal Conductance

The analysis of $A_n$ showed that there were statistical differences for leaf age ($p \leq 0.001$), but not differences between the two dates, 11 October and 22 December of 2017 ($p > 0.05$).

The $A_n$ decreased as leaf age increased, with the exception of leaf 8 (the youngest leaf), which had a lower net photosynthetic rate than leaf 7. Stomatal conductance ($G_s$) declined significantly with increasing leaf age ($p \leq 0.05$; Table 2). There was no significant interaction between the date and Gs ($p > 0.05$).

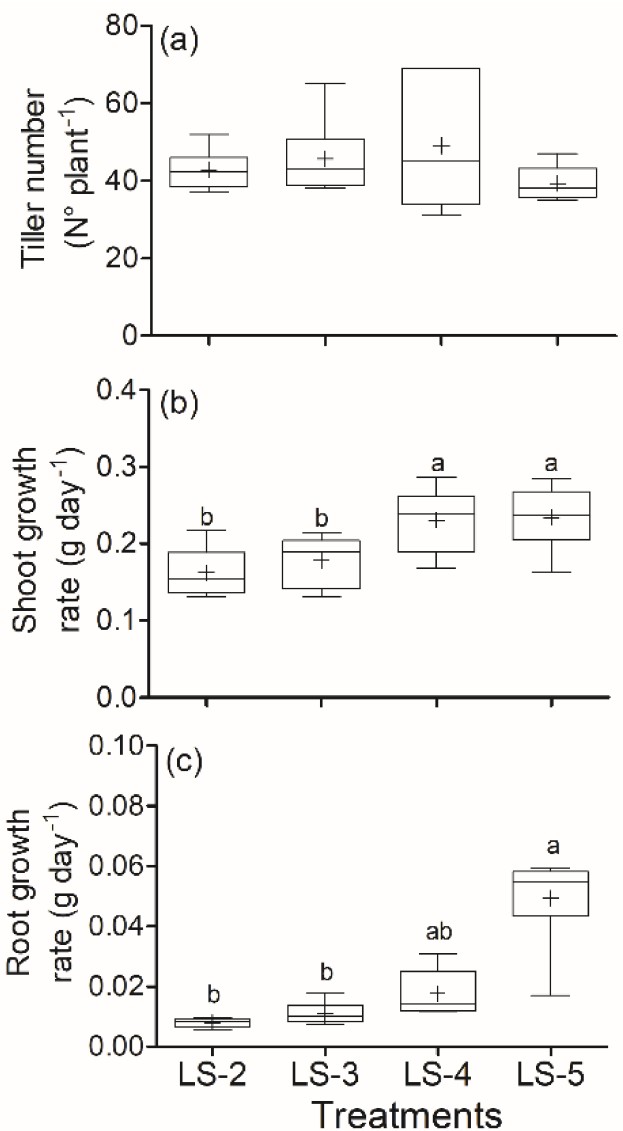

**Figure 4.** Tiller number (**a**), shoot growth rate (**b**) and root growth rate (**c**) for *B. valdivianus* defoliated by leaf stage (LS). The defoliation frequencies were based on defoliation at leaf stage 2 (LS-2); leaf stage 3 (LS-3); leaf stage 4 (LS-4); leaf stage 5 (LS-5). Lowercase letters indicate differences between treatments. + indicates the mean, while bars and boxes are the distribution of the population (quartiles) (*n* = 12).

### 3.4. Canonical Variate Analysis

The canonical variate analysis (CVA) showed the relationships between the measured variables and the defoliation regimes (Figure 6). The first two canonical variates explained 98% of the differences between treatments (Wilk's Lambda: $p \leq 0.001$). The first canonical variate (CAN 1; $p \leq 0.001$) explained 74% of the differences between the treatments and the second canonical variate (CAN 2; $p \leq 0.001$) explained 24%.

Figure 6a shows that the longer defoliation treatments, LS-4 and LS-5, were more positively associated with most of the parameters related to plant growth (Figure 6b) than

LS-2 and LS-3. Root growth rate and total sugars were more related to LS-5 than LS-4. All the positive plant parameters correlated with CAN 1 were more related to LS-4 than LS-5.

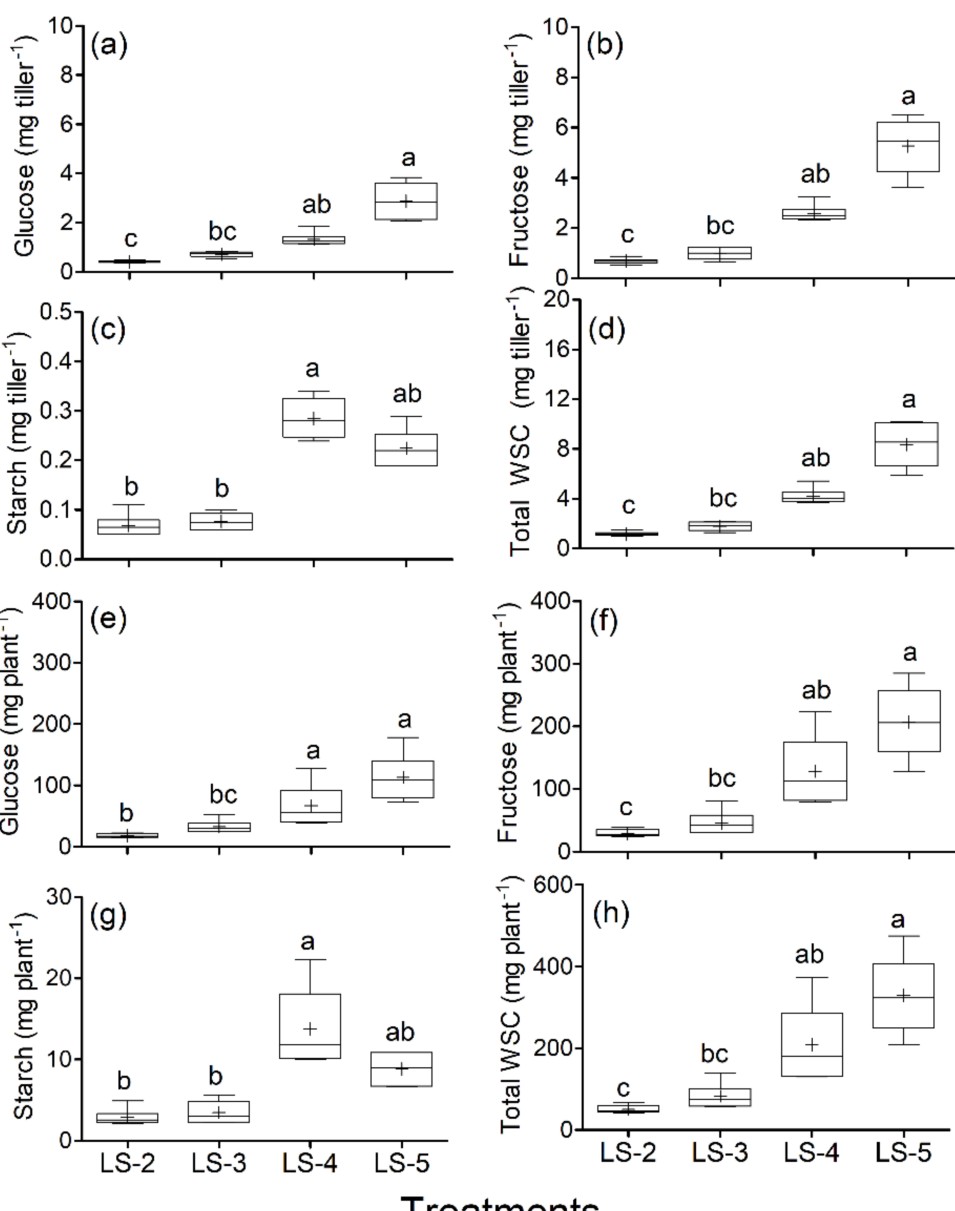

**Figure 5.** Water-soluble carbohydrates: mg of glucose per tiller (**a**), mg of fructose per tiller (**b**), mg of starch per tiller (**c**), mg of total WSCs per tiller (**d**), mg of glucose per plant (**e**), mg of fructose per plant (**f**), mg of starch per plant (**g**) and mg of total WSCs per plant (**h**) for *B. valdivianus* defoliated by leaf stage (LS). The defoliation frequencies were based on defoliation at leaf stage 2 (LS-2); leaf stage 3 (LS-3); leaf stage 4 (LS-4); leaf stage 5 (LS-5). The content per plant was determined by multiplying the quantity of the WSCs per tiller by the average tiller population per plant determined at the end of the study. Lowercase letters indicate difference between treatments. + indicates the mean, while bars and boxes are the distribution of the population (quartiles) (*n* = 17).

Figure 6b shows that the accumulated herbage mass, lamina area growth rate, tiller number and shoot growth rate were positively associated with CAN 1, while the root growth rate was negatively associated. Accumulated herbage mass, plant lamina weight, tiller weight, shoot growth rate, root growth rate and total sugars (WSC + starch) were

positively associated with CAN 2, but no plant parameter was positively associated with CAN 2 in its negative direction.

**Table 2.** Net photosynthesis ($A_n$) and stomatal conductance ($G_s$) for *B. valdivianus* during two evaluation dates at different leave ages.

| | 11 October 2017 | | 22 December 2017 | |
|---|---|---|---|---|
| **Leaf Number** | $A_n$ ($\mu$mol cm$^2$ s$^{-1}$) | $G_s$ ($\mu$mol cm$^2$ s$^{-1}$) | $A_n$ ($\mu$mol cm$^2$ s$^{-1}$) | $G_s$ ($\mu$mol cm$^2$ s$^{-1}$) |
| Leaf 4 | 7.6 c | 0.04 c | 7.0 c | 0.09 c |
| Leaf 5 | 9.5 b | 0.05 b | 9.3 b | 0.13 b |
| Leaf 6 | 10.9 a | 0.08 a | 11.4 a | 0.19 a |
| Leaf 7 | 11.00 a | 0.09 a | 11.8 a | 0.20 a |
| Leaf 8 | 9.0 b | 0.10 a | 10.4 b | 0.16 a |
| *p*-value | *** | * | *** | * |

$A_n$ and $G_s$ correspond to the last five expanded leaves for the LS-5 treatment (defoliation at leaf stage 5). Leaves are from two tillers per pot and six pots were evaluated. SEM is the standard error of the mean (*n* = 6). The order is from the oldest (leaf 4) to the youngest leaf (leaf 8). *** means $p \leq 0.0001$; * means $p \leq 0.05$.

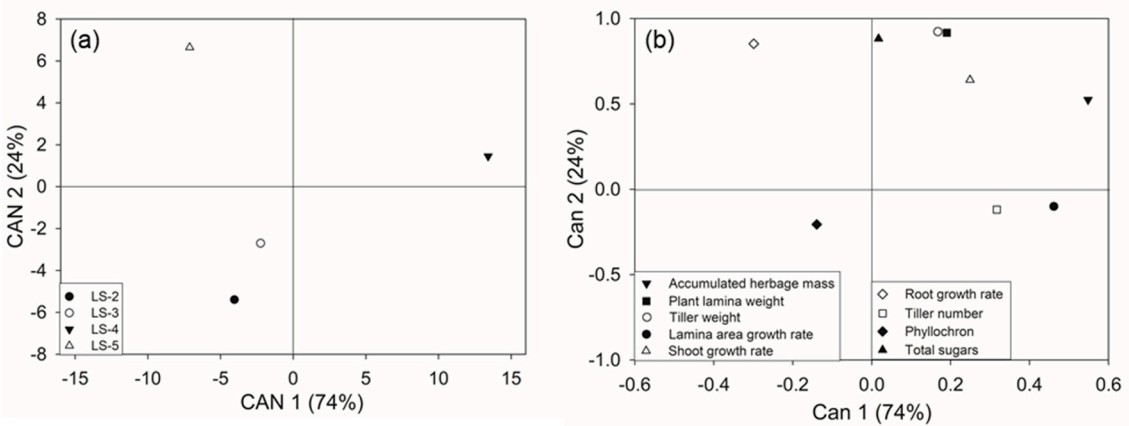

**Figure 6.** (**a**) Differences between each defoliation frequency treatment. The defoliation frequencies were based on defoliation at leaf stage 2 (LS-2; full circle); leaf stage 3 (LS-3; empty circle); leaf stage 4 (LS-4; full triangle); leaf stage 5 (LS-5; empty triangle). (**b**) Canonical variates showing the effect of defoliation frequency based on the plant leaf stage development for *B. valdivianus* evaluated through plant growth and physiological parameters.

## 4. Discussion

The current study determined that *B. valdivianus* maintains six live leaves per tiller$^{-1}$, with three of these fully expanded, and the remaining three simultaneously expanding. The beginning of senescence was at 4.25 LS, that is, when the seventh leaf began its growth and the second leaf began its senescence. It was considered that the second leaf was the first relevant leaf that showed major growth, and therefore made a relevant contribution to plant photosynthesis in comparison to the first leaf. No differences were found in relation to the phyllochron and tiller population. When *B. valdivianus* reached LS-4 development, *L. perenne* reached LS-3.5 development. At LS-4 onwards, *B. valdivianus* showed a higher shoot growth rate and leaf expansion rate. However, at LS-5, *B. valdivianus* reached a higher root growth rate, tiller leaf area, tiller weight and WSC accumulation. However, no statistical differences were obtained for the shoot growth rate, root growth rate or WSC accumulation when LS-5 was compared to the LS-4 treatment.

### 4.1. Leaf Growth Dynamics

The leaf growth dynamics represent a major growth difference between *B. valdivianus* and *L. perenne*, as *L. perenne* only has one expanding leaf at a time and maintains a total of three live leaves per tiller$^{-1}$ [16,28]. *Dactylis glomerata* L. (*D. glomerata*) and *Bromus*

*willdenowii* Kunth. (*B. willdenowii*) maintain five live leaves per tiller$^{-1}$ and grow two leaves simultaneously, with the first leaf reaching 75% of its total expansion length before the next leaf appears [36]. For *B. valdivianus*, when the first leaf reached its maximum expansion, the second leaf had reached 75% of its maximum expansion, the third leaf had reached 25% of its expansion and the fourth leaf was starting to grow. Leaf senescence began with leaf number 2 and started when tillers had accumulated 7 live leaves (i.e., LS-4.25).

On average, a new *B. valdivianus* leaf appeared every 96 °C d and each individual leaf required 431 °C d for full expansion, with three leaves expanding simultaneously. These results contrasted with leaf parameters determined for other temperate pasture grass species, such as *L. perenne,* in which a new leaf appeared every 117 °C d [37]. The strategy of *B. valdivianus* appears to be focused on leaf number, tiller and leaf size more than tillering itself, a major difference in growth strategy in comparison with *L. perenne.* It has been reported that *L. perenne* grown in a subtropical environment and defoliated with three live leaves at 5 cm height had a tiller weight of 0.036 g, and plants averaged 86 tillers [17]. *Lolium perenne,* when defoliated to 6 cm height, had four times more tillers than *B. willdenowii* [38]. These results for *L. perenne* contrasted with those of *B. valdivianus,* which reached a tiller population of 41 tillers plant$^{-1}$, tiller weight of 0.134–0.6 g and leaf area of 12.7–44.3 cm$^2$. The current study described the growth strategy of *B. valdivianus* in terms of the number of live leaves maintained per tiller, the size and the lifespan of individual leaves and the low tiller number per plant, which are all different strategies to those exhibited by *L. perenne.* These strategies have all been successful in the south of Chile on Andisol soils, where both species coexist in the same pasture ecosystems, with both *B. valdivianus* and *L. perenne* strongly contributing to the pasture production [1,11,13].

### 4.2. Net Photosynthesis and Stomatal Conductance Related to Leaf Ageing

The average $A_n$ for the five measured leaves on *B. valdivianus* was 17.15 μmol $CO_2$ m$^2$ s$^{-1}$, and with the leaf area per tiller at LS-4 being 24.8 cm$^2$, the total $A_n$ per tiller was 0.0425 μmol s$^{-1}$. This value is 8 times greater than *L. perenne,* assuming a leaf area of approximately 2.7 cm$^2$ tiller$^{-1}$ at LS-3 [38], and an average $A_n$ of 20 μmol $CO_2$ m$^2$ s$^{-1}$ [39], giving total photosynthesis per tiller of 0.0054 μmol s$^{-1}$. This greater $A_n$ per tiller for *B. valdivianus* over *L. perenne* could be one of the advantages of *B. valdivianus* in a mixed sward with *L. perenne.* For example, *B. valdivianus* could allocate more photoassimilates to root growth, resulting in higher survival and growth rates during summer in comparison to *L. perenne* [13], in addition to higher water extraction from deeper soil layers [7].

The decrease in both $A_n$ and $G_s$ as leaf age increased was also reported for the leaves of potato [40] and maize [41]. The decrease in photosynthetic capacity has been related to several causes, such as the electron transport rate [41], chlorophyll content [42] and carotenoids [43] due to the lipid peroxidation induction by reactive oxygen species (ROS) [44].

### 4.3. Effect of Defoliation Frequency on Water-Soluble Carbohydrates and Starch Reserves

The current study supports the idea that defoliation frequency has a direct impact on the storage of WSCs [18,19,26,30,45] and on subsequent plant regrowth (the leaf elongation rate) within the first three days after defoliation [26,46] on *B. valdivianus.*

The WSCs and starch vary with day/night temperature, amount of daylight, levels of nutrients and water availability, which act to alter the relationship between respiration and photosynthesis [16]. Many researchers have investigated the defoliation interval and its effect on WSC storage and subsequent regrowth and survival of various grass species [17,26,28,46]. Above 160 g WSC kg$^{-1}$, there is no longer an effect on the regrowth of *L. perenne* [26]. For *D. glomerata,* defoliation at LS-4 was recommended, with a content of 218 mg WSC plant$^{-1}$, equivalent to 236 g WSC kg$^{-1}$ [19], and furthermore for *D. glomerata,* a defoliation frequency between LS-2 and LS-4 was recommended, which corresponded to 70 mg WSC plant$^{-1}$ and 1477 mg WSC plant$^{-1}$, respectively [45]. For *B. willdenowii,* defoliation at LS-4 was recommended, with a WSC content of approximately

600 mg plant$^{-1}$ [47]. These results indicate how difficult an 'optimum' level of WSC is to define, especially when the grass species have different growth strategies. According to the current study, WSCs reached concentrations around 60 g kg$^{-1}$ through LS-2 to LS-4, increasing to 140 g WSC kg$^{-1}$ at LS-5. However, several authors have reported that WSC content (expressed as either mg tiller$^{-1}$ or mg plant$^{-1}$) is more significant than the WSC concentration (expressed as % or g kg$^{-1}$) in regrowth and survival (e.g., [17,25,45]). The latter is supported by the present study, where the WSC content increased as a response to more infrequent defoliation, together with the growth plant parameters and regrowth. In this study, LS-4 and LS-5 treatments were the defoliation frequencies that encouraged the accumulation of energy reserves. The increasing rates of WSC accumulation between the different defoliation frequencies, i.e., the WSC median for LS-2 and LS-3, were 50 and 82 mg plant$^{-1}$, respectively, and for LS-4 and LS-5 they were 209 and 329 mg plant$^{-1}$, respectively (Figures 5 and A1). The priorities for the allocation of energy reserves are as follows: (1) Restoration of the photosynthetic capacity with the development of new leaves; (2) replenishment of the WSCs in the storage organs (stubble); (3) tiller initiation; and (4) root growth [16]. Therefore, it can be hypothesised that *B. valdivianus* restored its photosynthetic capacity around LS-4 when it changed the priority for the allocation of photoassimilates by increasing the allocation towards energy reserve restoration and root growth.

It is recognised that starch is the main storage product of many seeds, e.g., grasses [48]. The seeds initially develop in the stubble of the grasses, which explains the constant increase in starch until treatment at LS-4, as the seeds were being developed from the first 5 cm of the plant. The LS-5 starch concentration diminishment was due to its movement from the stubble to the inflorescence, which occurred for that treatment.

### 4.4. Effect of Defoliation Frequency on Plant Regrowth

In the current study, although there was no statistical difference in tiller number ($p > 0.05$), at LS-5, *B. valdivianus* had a greater tiller leaf area, tiller weight and tiller leaf weight than all other treatments. When leaf expansion was analysed, the derivative of the growth curve on the third day of the growing period showed that LS-4 had a steeper slope than LS-5. The higher leaf expansion rate of LS-4 and LS-5 could be the effect of greater WSC accumulation, allowing the plant to restore its photosynthetic tissue faster than the plants with a deficient WSC accumulation (i.e., LS-2 and LS.3). The shoot growth rate at LS-5 was similar to LS-4 and resulted in the maximum growth rate as shown in Figure 4b, as any earlier defoliation will negatively affect the biomass production [38] and plant persistence [16].

A decrease in WSCs when defoliation frequency increased to LS-2 or LS-3 for *B. valdivianus* indicated a detrimental effect on plant recovery and the plant shoot growth rate, root growth rate, tiller leaf weight and tiller leaf area. A similar effect of the defoliation frequency on WSC levels has been reported for *L. perenne* [16]. It is important to highlight that LS-4 did not show a detrimental effect on WSC content (mg plant$^{-1}$ or mg tiller$^{-1}$) and was statistically similar to LS-5. These results are also supported in the CVA, which showed most of the plant parameters that were evaluated were highly related to LS-4 and LS-5 (CAN 1). The total sugars (in CAN 2) showed a high correlation with LS-5 followed by LS-4.

The impact of the frequent defoliations, besides the depletion of WSCs in the stubble, resulted in a detrimental effect on the regrowth in leaves, roots and tillers, which may negatively affect plant persistence within the pasture [25,26]. Contrarily, tiller number was not affected by the defoliation regimes, whereas the root growth rate, shoot growth rate and WSC accumulation were affected by defoliation regimes (Figure A1). These results raise doubts about whether the prioritisations of the photoassimilates indicated above (i.e., leaf growth, energy restoration, tiller initiation, root growth) [16] is the same for *B. valdivianus.* Because no differences related to tiller population were found, it appears that the leaf growth followed by WSC and starch accumulation and root growth were the

three main destinations for the photoassimilates generated by *B. valdivianus*. From this, it could be hypothesised for *B. valdivianus* that (1) allocation patterns can be managed by the defoliation criteria applied and (2) allocation patterns are different in comparison to *L. perenne*.

## 5. Conclusions

*Bromus valdivianus* is a 'six-leaf' species, with three leaves growing simultaneously, and leaf senescence beginning at LS-4.25, with leaf appearance every 96 °C d and 431 °C d required for a full lamina expansion.

The lower defoliation frequencies LS-4 and LS-5 encouraged WSC and starch accumulation, shoot regrowth (photosynthetic tissue) and root growth, but no effect was found related to the tiller population under the different defoliation frequencies. As the tiller population was the same for all treatments, the greater plant regrowth shown by LS-4 and LS-5 was due to their greater tiller size, tiller leaf area and leaf elongation rates.

The higher growth rates reached by LS-4 and LS-5 were related to higher WSC accumulation in the stubble. Furthermore, LS-4 and LS-5 treatments showed an increase in their median values of WSC accumulation rates and root growth in comparison to LS-2 and LS-3, indicating a switch in the plant prioritisation of resources from LS-4 onwards.

The LS-4 stage in *B. valdivianus* is homologous to the LS-3.5 stage for *L. perenne*, indicating a longer period is needed between defoliations to reach an appropriate amount of WSC accumulation and to encourage the plant regrowth for *B. valdivianus* when compared to the results from the vast literature related to the defoliation frequency studies on *L. perenne*.

The net photosynthetic rate and stomatal conductance of *Bromus valdivianus* were not significantly different to that reported for other grasses (e.g., *L. perenne*), but its advantage, from the tiller point of view, was the higher leaf area, generating higher total photoassimilates per tiller.

**Author Contributions:** I.P.O.: Performed the conceptualisation, methodology, software analysis, validation of the data, formal analysis, investigation, data curation and collection, writing—original draft preparation, writing—review and editing, and visualisation. I.F.L.: Main supervisor involved in the conceptualisation, methodology, software analysis, validation of the data, formal analysis, investigation, data curation and collection, writing—original draft preparation, writing—review and editing and visualisation, supervision, project administration, funding acquisition and resources. P.D.K.: Supervisor involved in the conceptualisation, methodology, software analysis, validation of the data, formal analysis, investigation, writing—original draft preparation, writing—review and editing and visualisation, supervision, project administration, funding acquisition and resources. D.J.D.: Supervisor involved in the conceptualisation, methodology, validation of the data, formal analysis, investigation, writing—original draft preparation, writing—review and editing and visualisation, supervision, project administration, funding acquisition and resources. Y.Z.: Researcher involve in software analysis, validation of the data, writing—review and editing, visualisation and data curation. P.H.: Specifically in data collection and investigation. All authors have read and agreed to the published version of the manuscript.

**Funding:** This work was supported by the Massey University Research Fund (MURF) and the School of Agriculture and Environment.

**Institutional Review Board Statement:** Not applicable.

**Informed Consent Statement:** Not applicable.

**Data Availability Statement:** The results presented in this study are available on request from the corresponding author.

**Acknowledgments:** The authors thank the Massey University Research Fund (MURF) for funding this study. Thanks to NZ Agriseeds Ltd. for providing the seeds used in the study. Thanks to Cory Matthew ([1]), Mark Osborne ([1]), Soledad Navarrete ([1]), Sunmeet Bhatia ([1]) and Felipe Hernández ([2]) for their assistance during the development of this study. ([1]) School of Agriculture, Massey University, New Zealand. ([2]) Facultad de Ciencias Agrarias y Alimentarias, Universidad Austral de Chile, Chile.

**Conflicts of Interest:** The authors declare no conflict of interest. The funders had no role in the design of the study; in the collection, analyses, or interpretation of data; in the writing of the manuscript, nor in the decision to publish the results.

**Appendix A**

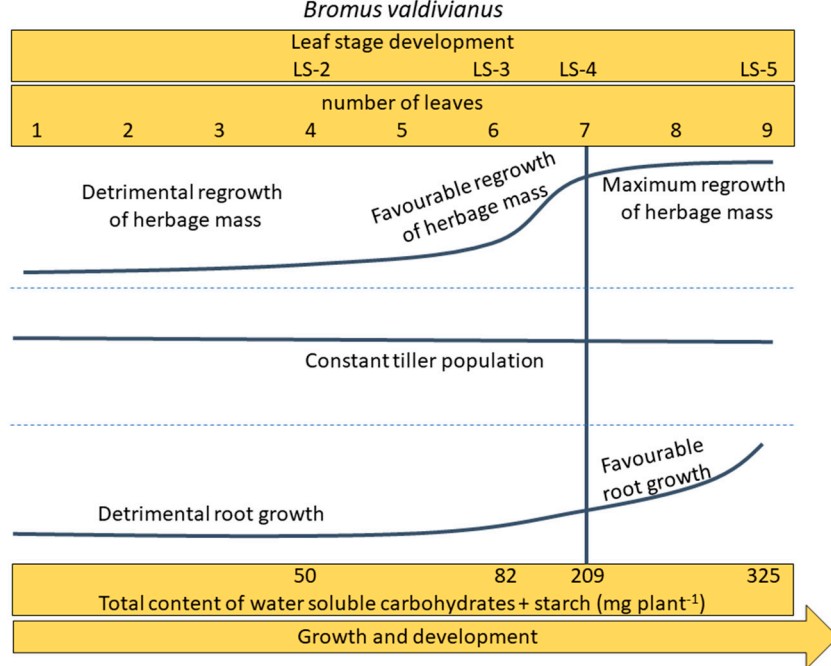

**Figure A1.** Effect of defoliation frequency, based on leaf stage development or number of leaves, on *Bromus valdivianus* total accumulation of non-structural carbohydrate reserves (water-soluble carbohydrates + starch) and the following repercussions on regrowth, evaluated through tiller population, herbage mass and root growth.

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
