# Peer review of "Response of Bromus valdivianus (Pasture Brome) Growth and Physiology to Defoliation Frequency Based on Leaf Stage Development"

_agronomy, doi:10.3390/agronomy11102058_

Round 1
Reviewer 1 Report
Comments to Editor:
The authors investigated the “Response of bromus valdivianus (pasture brome) growth and physiology to defoliation frequency based on leaf stage development.” However, the authors are advised to add more relevant literature in introducing the effects management of defoliation, and discuss its impact on bromus physiology and dry matter accumulation under the prevailing conditions. Then, the message of this manuscript will be more effective and clearer for readers. Based on my comments below, I would recommend a major revision of the manuscript.
Specific Comments
Abstract
The abstract needs to be revised (e. g., give details of main results and link your results with treatment and their effects on bromus; name the methods to achieve the goal/aim described in the background).
Introduction
The introduction is not providing the enough information on the research background and research gap. Authors are not very clear about what problem they are going to solve. Moreover, an introduction always needs to be very clear on what we already know, what we don’t know, and which questions are therefore addressed by the research. The authors are also advised to add some more information on the effects of defoliation on WSC, LS and Photosynthetic parameters.
Materials and Methods
The authors are advised to add weather data to their experimental site because changes in weather conditions significantly affect plant growth and development. Although it is a pot study but still this weather data is relevant to the research question for broader picture.
Results
I recommend the authors calculate each treatment's gross economic profit to determine the optimized defoliation level for bromus production.
Discussion
I think authors should rethink what they write in the first paragraph, and only summarize the main findings in view of the research questions. After this, authors can explore in subsequent paragraphs different aspects of the work and explain how their findings expand the envelope of knowledge, but first of all, authors simply need to state the main results without discussing their why and how or the relationships to the literature. First of all, the reader needs a clear statement on what the study found. Moreover, it is suggested to discuss the main results in a logical way.
Reviewer 2 Report
Generally very well written. I suggest deleting the first sentence of the paragraph starting line 420. It talks about NSC and the rest of the paragraph is about WSC.
Reviewer 3 Report
Main comment:
The manuscript presents the effect of Bromus valdivianus defoliation at different stages of vegetative growth on growth dynamics of leaves and roots, photosynthesis and content of water soluble carbohydrates and starch accumulation.
The research was carried out correctly and the descriptions in the methodology are quite accurate, and in some places they need to be detailed, primarily concerning- heat units, expressed in growing degree days (GDD). The manuscript used the term: accumulated growing degree days or total of growing days, while in the description of the figures the term: growing degree days. It also did not explain why the temperature was based on 5°C for the calculation of this parameter (Line 148-150). There is also no mention of growing degree days in the title or abstract and keywords.
Other comments and suggestions:
Line 2: Bromus valdivianus (italic)
Line 19: L. perenne physiological growth- does not specify a concept- rather: morphological and physiological changes during growth or physiological growth and development
Line 24: Instead of gas exchange (there was no transpiration measured): photosynthetic rate and stomatal conductance
Line 25: tiller number per plant
Line 34: Keywords, also: growing degree days; photosynthetic rate; stomatal conductance
Line 42: 1.543 MPa
Line 77: The aim was to determine B. valdivianus growth dynamics and the effects that defoliation at different stages of vegetative development have on B. valdivianus growth (not only the growth) This sentence is not precise.
Line 145: GDD? (growing degree days)
Line 180: photosynthetic rate (An) and stomatal conductance (Gs)
Line 186: at 1000 μmol photons
Line 202: growing degree days (GDD),
Line 234, 245 and 255: Bromus valdivianus (italic)
Line 270: Figure 3: too large font next to axes
